# Therapeutic Effect of Mesenchymal Stem Cells in Ulcerative Colitis: A Review on Achievements and Challenges

**DOI:** 10.3390/jcm9123922

**Published:** 2020-12-03

**Authors:** Seyed-Kazem Hosseini-Asl, Davood Mehrabani, Feridoun Karimi-Busheri

**Affiliations:** 1Department of Internal Medicine, School of Medicine, Shiraz University of Medical Sciences, Shiraz, Fars 71348-14336, Iran; gehrc@sums.ac.ir; 2Stem Cell Technology Research Center, Shiraz University of Medical Sciences, Shiraz, Fars 71348-14336, Iran; 3Burn and Wound Healing Research Center, Shiraz University of Medical Sciences, Shiraz, Fars 71987-74731, Iran; 4Comparative and Experimental Medicine Center, Shiraz University of Medical Sciences, Shiraz, Fars 71348-14336, Iran; 5Department of Oncology, Faculty of Medicine, University of Alberta, Edmonton, AB T6G 1Z2, Canada

**Keywords:** inflammatory bowel disease, ulcerative colitis, therapy, mesenchymal stem cells

## Abstract

The worldwide epidemiology of inflammatory bowel disease (IBD), including Crohn’s disease (CD) and ulcerative colitis (UC), still shows an increasing trend in Asia and Iran. Despite an improvement in the treatment landscape focused on symptomatic control, long-term colectomies have not decreased over the last 10-year period. Thus, novel therapies are urgently needed in clinics to supplement the existing treatments. Mesenchymal stem cells (MSCs) are multipotent adult stem cells with immunosuppressive effects, targeting IBD as a new treatment strategy. They have recently received global attention for their use in cell transplantation due to their easy expansion and wide range of activities to be engrafted, and because they are home to the mucosa of the intestine. Moreover, MSCs are able to differentiate into epithelial and other cells that can directly promote repair in the mucosal damages in UC. It seems that there is a need to deepen our understanding to target MSCs as a promising treatment option for UC patients who are refractory to conventional therapies. Here, we overviewed the therapeutic effects of MSCs in UC and discussed the achievements and challenges in the cell transplantation of UC.

## 1. Introduction

Dynamic interactions between the gastrointestinal epithelium and the mucosal immune system result in intestinal homeostasis and optimal immunosurveillance, but a destabilization in these interactions among predisposed people can lead to development of chronic inflammatory diseases called inflammatory bowel disease (IBD), which mainly covers Crohn’s disease (CD) and ulcerative colitis (UC) [1]. UC affects the rectum and may involve any part of the colon in a continuous pattern, whereas CD can affect any part of the gastrointestinal tract [2]. UC has a remitting and relapsing course that can vary from asymptomatic mild inflammation to extensive inflammation of the colon leading to colonic motility dysfunction, frequent bloody stools, potentially permanent fibrosis, and tissue damages [2]. It is usually characterized by abdominal pain, fever, weight loss, diarrhea, rectal bleeding and atrophy, and intestinal obstruction. It has an easy relapse that may result in surgical options to remove the affected bowel [3].

UC has also several extra-intestinal manifestations in eyes, skin, joints, and the oral cavity and is associated with hepatobiliary diseases, osteoporosis, and amyloidosis [3]. Histologically, the lesions in UC are mostly inflammatory and ulceration limited to the mucosa and submucosa layers in the colon and rectum [4]. UC is associated with secretion of a variety of chemokines and cytokines, including interleukin (IL) 1β, IL-6, IL17A, and IL-21, Gro-α, macrophage inflammatory protein (MIP)-1α, MIP-2, eotaxin, tumor necrosis factor α (TNF-α), and reactive oxygen species (ROS) [5].

### 1.1. Epidemiology of UC

An episode of acute severe UC was shown to be a life-threatening condition among one-fourth of patients [6]. Approximately 31% of patients with limited UC at diagnosis would have disease extension by 10 years [7], while 74% of them may experience at least one relapse during a 5-year period [8]. In 10–15% of patients, the disease can ultimately lead to colectomy [6]. Both male and female individuals are affected equally, and the adult population aged 30–40 years old are affected more [6], although younger patients between 15 and 25 years old may experience a flare up too [6].

It is known that these patients are prone to increased risks of developing colorectal cancer (CRC), which has global incidence and mortality rates of 10.2% and 9.2%, respectively [9], with an increasing trend in newly industrialized countries in Asia and Latin America [10]. The overall incidence rate for UC is coalescing around 5–15 per 100,000 person-years, which varies depending on geography [10]. The highest annual incidence was reported 6.3 per 100,000 person-years in Asia, while in the Middle East and in North America and Europe approximately 10–200 cases per 100,000 individuals were reported [11].

### 1.2. The Etiology and Pathogenesis of UC

Though the etiology of UC has not been defined clearly, the disease is believed to be multi-etiological. It is driven by immune dysregulation of mucosa toward luminal bacterial floral, an immune imbalance between regulatory and effector T cells, and the inflammatory cascade induced by leukocyte recruitment, infection, genetic alterations, and environmental factors such as diet, lifestyle, and socioeconomic development that lead to destruction of the epithelial barrier. Recent studies indicated both innate and adaptive immunity to play a part in disease pathogenesis [12]. Various factors, such as inflammatory reaction, tight junction epithelial barrier dysfunction, and apoptosis are also associated with the disease [13].

### 1.3. The Medical Therapy of UC

The goals in the medical therapy of UC are to ameliorate major symptoms of the disease, to treat extraintestinal manifestations and to prevent complications. Clinically, treatment is mainly focused on anti-inflammatory properties, mucosal repair, management of risking future relapse, and protection to lower the risk of requiring colectomy [6], because a colectomy can be associated with complications in one-third of patients [14]. The therapies administered during the early course of the disease were shown to modify the disease progression [15]; still, despite an improvement in the treatment landscape, long-term colectomy rates have not seriously decreased over a 10-year period [6].

Current treatment of UC primarily consists of 5-aminosalicylates, glucocorticoid (GC), and antimicrobials, as well as immunomodulators such as azathioprine, 6-mercaptopurine (6-MP), cyclosporine, methotrexate, and thiopurines, and biologics such as monoclonal antibodies and anti-tumor necrosis factor (TNF)-α, but these treatment methods may cause side effects and can lead to treatment resistance and remission [16]. The use of corticosteroids was demonstrated to be associated with weight gain, hyperglycemia, cutaneous side effects, osteoporosis, adrenal insufficiency, and an increase in the risk of opportunistic infections, especially when administered in combination with other immunosuppressive drugs that may not be well tolerated—these side-effects cause nearly one-fourth of patients to discontinue their treatments [16].

Treatment resistance may need surgical intervention, and can lead to a total colectomy that can severely compromise the quality of life in UC patients [6]. A meta-analysis based on seven population-based studies published between 2008 and 2013 regarding 5140 UC patients showed that post-operative mortality of emergency colectomy for UC was 5.3% [17], and the risk of carcinogenesis should also not be ignored [9]. Thus, promising new strategies and safe treatments for UC are urgently needed to improve the control of the disease, such as emerging therapies with mesenchymal stem cells (MSCs) [18]. In our review, a comprehensive literature review was undertaken in August 2020 across Medline, Embase, Google Scholar, and ClinicalTrials.gov databases on the achievements and challenges in the treatment of UC using MSCs. Keywords used were “Inflammatory bowel diseases,” “Mesenchymal stem cells,” and “Ulcerative colitis.” The findings were categorized based on clinical trials and in vivo studies and further based on the type of MSCs used. The animal model and the method of colitis induction were described too.

### 1.4. The Characteristics of MSCs as Emerging Therapy of UC

MSCs were primarily recorded as a group of non-hematopoietic, self-renewing, plastic-adherent and fibroblast-like stromal cells [19] that have the ability to transdifferentiate into ectodermal and endodermal cells (e.g., chondrogenic, adipogenic, and osteogenic) [20]. They express cell surface markers such as CD44, CD73, CD90, and CD105, and lack the expression of hematopoietic markers such as CD34 and CD45 [21]. They have been isolated from various sources such as bone marrow (BM) [22], adipose tissue [23], and umbilical cord Wharton’s jelly [24]. MSCs have therapeutic effects in various inflammatory diseases due to its hypoimmunogenic and immunoregulatory properties and can be the home to the site of injury, limit inflammation through cytokine release, stimulate healing through growth factor expression, alter host immune responses through secretion of immune-modulatory proteins, and secrete anti-apoptotic factors [25].

### 1.5. The Modulating Role of MSCs in UC

In the IBD microenvironment, there is an imbalance of T cell subsets and Bax (proapoptotic) and Bcl-2 proteins (antiapoptotic), which are responsible for defective immune cell apoptosis [26]. The migration of MSCs to the injury sites can promote the tissue repair and modulate the function of immunocytes, such as T cells, B cells, NK cell, dendritic cells (DCs), and macrophages by modulating their proliferation and differentiation, suggesting that the therapeutic action is unlikely to be due to replacement of diseased tissue [27]. Macrophages are the most important cells for the induction of colon inflammation [28]. It was shown that MSCs enhance their therapeutic effects by maintaining a balance between M1 and M2 macrophages, while MSCs can secrete tumor necrosis factor-α-induced gene/protein 6 (TSG-6) and induce a macrophage phenotypic switch from M1 to M2 [29].

One major role of MSC cytokine regulatory repair in UC is to restore the lost balance between Treg cells and proinflammatory Th1/Th17 cells to recruit circulating leucocytes and to stimulate the macrophages and B cells in the intestine. MSCs can effectively harbor the injured tissue and secrete powerful immunoregulatory soluble factors to inhibit the proliferation and function of Th1/Th17 cells and to promote Treg differentiation together with survival and recovery of injured cells and tissues. The outcome would be an increase in secretion of anti-inflammatory cytokines such as IL-4, IL-10, IL-11, IL-13, and TGF-β, and a decrease in inflammatory cytokines such as IL-6, IL-12, IL-21, IL-23, and NF-κB activities [30,31].

MSCs can effectively induce T cell apoptosis via the FAS ligand (FASL)-dependent pathway to attenuate UC. In this mechanism, FAS-modulated monocyte chemotactic protein 1 (MCP-1) secreted by MSCs can recruit the T cells for FASL-mediated apoptosis, while the apoptotic T cells further stimulate macrophages to express high levels of TGFβ leading to upregulation of CD4(+)CD25(+)Foxp3(+) Treg cells and, finally, an immune tolerance that can accelerate the recovery of colitis [32,33]. MSCs have direct inhibitory functions on the antigen-presenting activities of macrophages and dendritic cells, making them immunologically tolerant by an increase in the secretion of IL-10 and heighten the induction of Treg in UC [34]. MSCs can also attenuate IBD through the reduction in expression of 15-lox-1 in macrophages that results in enhanced colonic tissue repair [35].

### 1.6. The Therapeutic Role of MSCs in UC

MSCs have received worldwide attention in regenerative medicine and cell therapy due to their easy expansion and wide range of activities [21]. Currently, over 400 clinical trials are on-going, including 26 clinical trials registered for IBDs, of which 26 are investigating the use of MSCs in CD and UC [36]. The origin and type (autologous or allogeneic) of MSCs, their preparation quality control to ensure suitable cellular differentiation in target locations, and the method of administration (route, dosage, schedule, pretreatment with chemokines or cytokines, etc.) can affect the final outcome of the treatment of UC [37]. MSCs could be easily isolated and are amplified from several tissues for the treatment of UC, including bone marrow-derived stem cells (BMSCs) [38], adipose tissue-derived stem cells (adSCs) [29], umbilical cord stem cells (UCSCs) [39], endometrial regenerative cells (EndSCs) [40], placenta [41], intestinal stem cells (ISCs) [42], amnion [43], amniotic fluid [44], and tonsil [45]. Their routes of administration were reported to be intraperitoneal (IP), intravenous (IV), subcutaneous (SC), endoscopic, systemic, or anal injection, which can affect the outcome of cell transplantation in UC [46,47,48].

An impaired contribution of BMSCs was shown to be an important component of mechanisms in the incomplete/delayed healing of UC [49]. Shi et al., in a systematic review and meta-analysis of experimental and clinical studies in the treatment of UC, demonstrated MSCs to be an underlying method of treatment for the disease [50]. Exosomes derived from MSCs have also been used as new remedies in the therapy of UC through the delivery of mRNAs and miRNAs that can activate autophagy and inhibit apoptosis, necrosis, and oxidative stress in injured gut epithelial cells, also promoting the survival and regeneration [51], because MSCs exert protective functions and support the survival and regeneration of colonic epithelial cells and mucous barriers through the production of exosomes, growth factors, cytokines, and metabolites [52].

### 1.7. Undertaken Case Reports in the Treatment of UC by MSCs

Undertaken case reports in the treatment of UC by various sources of MSCs are presented in Table 1. In a case report, autologous BMSC transplantation as a suspension of nearly 4.0 × 10^8^ cells (volume: 50 mL) was completed for the treatment of UC at several locations through submucosal injection (2–4 mL per location). The findings showed alleviation of the patient’s symptoms; furthermore, the patients recovered from hematochezia, and there was a decrease in the patients’ hypersensitive C reactive protein values. Colonoscopy findings were associated with a decrease in lesions and bleeding and edema in the sigmoid colon and rectum without any recurrence in the subsequent two years [53]. In 2009, BMSCs were successfully used in patients suffering from UC as the first IV systemic cell transplantation [54]. Transplantation of BMSCs to the patients with UC has illustrated an improved clinical course, while the duration of remission showed an increase in chronic and continuous recurrent course of disease and the risk of relapse and the frequency of hospital admissions declined in comparison to conventional therapies [55,56]. In elderly patients with UC, the transplantation of BMSCs was less effective than in patients with young and middle age [57]. In BMSCs, aging was shown to be associated with decreased proliferative capacity and a significant decrease in differentiation properties with an increase in age [58].

### 1.8. Undertaken Clinical Trials in the Treatment of UC by MSCs

Undertaken clinical trials in the treatment of UC by various sources of MSCs are illustrated in Table 2. In a clinical trial phase II, a single intravenous (1.5–2) × 10^8^ dose of allogeneic BMSCs was used in the treatment of 44 patients with UC. After 24 months of follow-up, 72.7% of patients achieved the clinical and morphological indices of inflammatory activity [61]. The potential advantage of freshly isolated autologous AdSCs (Around 11 × 10^5^) in two UC patients was described before [62]. Among 25 UC patients treated with human peripheral blood mononuclear cells (PBMCs), it was shown that concomitant use of stem cells and IL-25 enhanced intestinal epithelial cell regeneration and treatment of the disease [63].

Hu et al. in 80 UC Phase I/II trials, using UCSCs (2.3–4.7 × 10^7^ cells/kg, 50 mL volume, IV), showed that the treatment group receiving stem cells had a >3 significant decrease Mayo score after 3 months in comparison to the placebo group [64]. The complete resolution of severe UC after haploidentical hematopoietic stem cell (HSC) transplantation and post-transplant high-dose cyclophosphamide was confirmed by endoscopic and histologic findings in a 44-year-old man [63]. In the case of allogeneic BMSC transplantation, it was shown that injected cells were safe and could contribute to clinical improvement in patients with refractory UC [65]. A total of 1 × 10^6^ FLK1+CD31-CD34-MSCs/kg as one IV injection in a Phase I clinical trial of IBD patients with a reduction in the extent of the inflamed area was reported by several researchers [65] (Table 2).

### 1.9. Registered Clinical Trials in the Treatment of UC by MSCs

There are several registered clinical trials for the treatment of UC, enrolling different sample sizes and various stem cell sources of BMSCs, AdSCs, UCSCs, WJSCs, ISCs, and HSCs. The dose and route of administration in cell transplantation methodology vary (Table 3).

### 1.10. In Vivo Studies Using BMSCs in the Treatment of UC

BMSCs have been used by several researchers in the treatment of experimental colitis using various animal models, including mice, rats, and Guinea pigs (Table 4). In these animals, some researchers have used trinitrobenzene sulfonic acid (TNBS) to induce colitis [37,65,66,67,68], while several studies reported the use of dextran sulfate sodium (DSS) to induce colitis models [47,48,68,69,70,71,72,73,74,75,76,77,78,79,80,81,82,83] and many authors applied acetic acid to induce colitis [1,3,4] (Table 4). The first BMSCs transplantation was conducted in 2006 in a TNBS-induced colitis mice model, revealing the important role of cell therapy in the repair of injured intestinal mucosa, as well as the downregulation of immune function of T cells [65]. Allogeneic transplantation of BMSCs in a TNBS-induced colitis rat model was demonstrated to populate the cells in the injured regions of the colon [66]. Intra-rectal administration of 1 × 10^6^ BMSCs in TNBS-induced colitis Guinea-pig model could prevent enteric neuropathic inflammation [67]. When the therapeutic effects of BMSCs and IFN-γ were compared in the treatment of a TNBS- and DSS-induced colitis mice models, it was shown that BMSCs had immunosuppressive effects and could enhance the capacity to inhibit Th1 inflammatory responses and diminish mucosal damage [68]. Administration of the conditioned medium derived from BMSCs in a TNBS- and DSS-induced colitis rat model led to production of pleiotropic gut trophic factors and an enhancement in the repair of intestinal epithelium [37] (Table 4).

Long-term repeated injections of BMSCs in a DSS-induced mice colitis demonstrated that cell transplantation at the onset of the disease exerted preventive and fast recovery effects with long-term immune-suppressive actions [47]. Endoscopic injection of BMSCs in a DSS-induced colitis mice model showed healing effects in colonic lesions [48]. Administration of BMSCs in a DSS-induced colitis rat model was shown to accumulate in inflamed tissues and ameliorate colonic lesions via a local anti-inflammatory action [69]. The abilities of BMSCs to effectively migrate and accumulate in inflamed regions of the colon were previously shown [69]. The differentiation of MSCs into vascular smooth muscle cells, endothelial cells, pericytes, or epithelial cells was illustrated to also protect colonic cells against apoptosis. The differentiation of BMSCs into colonic interstitial lineage cells could produce TGF-β1 and VEGF that could further participate in the healing of colitis lesions [69]. BMSCs in a DSS-induced colitis rat model were demonstrated to differentiate into several cells, to dampen the inflammation and to repair ulcerations and the distorted crypt architecture, and reduce the loss of surface columnar epithelium [70]. BMSC microvesicles, together with rebamipide in the DSS-induced colitis rat model, resulted in an improvement in the prompted colonic lesions [71]. BMSC transplantation in the DSS-induced colitis rat model could modestly promote the repair of colitis [72]. BMSC delivery through an IP route in a DSS-induced colitis mice model revealed higher proliferation of Ki-67 and FoxP3(+) cells, more BMSC migration to inflamed tissue, and an optimal mice colitis recovery in comparison to other routes of cell delivery [73]. BMSC transplantation in a DSS-induced colitis mice model appears to be a promising therapy to recruit macrophages and to produce TGFβ1 resulting in alleviation of colonic pathology [74]. Cell therapy of BMSCs in a DSS-induced colitis mice model showed a successful colonic mucosal regeneration [27]. Administration of BMSCs in a DSS-induced colitis mice model displayed beneficial effects in healing of colonic lesions due to suppressing inflammatory phenotype of dendritic cells in a galectin-3 (Gal-3)-dependent manner [75].

BMSCs and their extracellular vesicles (EVs) in a DSS-induced colitis mice model could attenuate the colitis injuries by promoting polarization of M2 macrophages [76,77]. Coating BMSCs with VCAM-1 in a DSS-induced colitis mice model resulted in an increase in cell migration to the inflamed colon and the promotion of tissue repair [78]. BMSC injection in a DSS-induced colitis mice model exhibited a reduction in the intestinal inflammation via the formation of aggregates, which consisted of macrophages and B and T cells, as well as immunomodulatory molecules such as FOXP3, IL-10, TGF-β, CCL22, heme oxygenase-1, arginase type II, and TSG6 within the peritoneal cavity of colitis mice. Subsequently, it exhibited an increase in Foxp3CD45+ cells and decrease in CD45+ cells, neutrophils, and metalloproteinase activities in the mucosa that could reduce the severity of colitis injuries [79].

Injection of BMSCs cultured in the presence of Galectin-3 (Gal-3) in a DSS-induced colitis mice model illustrated an increase in the sera concentration of IL-10 and an improvement in BMSC-mediated polarization towards the immunosuppressive M2 phenotype of macrophages [80]. The therapeutic effect of BMSCs was evaluated in a DSS-induced colitis mice model, and it was illustrated that IL-37b gene transfer enhanced the therapeutic efficacy of BMSCs [81]. Transplantation of BMSCs in a DSS-induced colitis mice model could ameliorate the oxidative stress and restore intestinal mucosal permeability [82]. A single dose of AdSCs in a DSS-induced colitis mice model showed significant reduction in the colitis disease activity index, a concomitant increase in the regulatory/inflammatory macrophage ratio in the colon lamina propria, and a sustained protection against acute inflammation in the long term [83].

In an acetic acid-induced colitis rat model, the anti-inflammatory and regulatory potency of BMSCs were noted to enhance the tissue regeneration in an injured gut. It was demonstrated that the migration of BMSCs into the reperfused small intestine reduced the oxidative stress based on the effects of superoxide dismutase, catalase, and glutathione peroxidase, as well as a reduction in the MDA level, which could ameliorate the clinical manifestations and tissue inflammation [46].

### 1.11. In Vivo Studies on Treatment of UC Using AdSCs

Regarding the use of AdSCs in the treatment of UC, rats and mice were used as experimental models of DS- and TNBS-induced colitis injuries (Table 5). In a TNBS-induced colitis rat model, the therapeutic potential of AdSCs was confirmed via the inhibition of inflammatory and autoimmune responses [84]. In a TNBS-induced colitis rat model, the immunoregulatory impacts of locally submucosal endoscopic injection of AdSCs was demonstrated via the expression of Foxp3 and IL-10 mRNA ameliorating colitis injuries with fewer inflammatory infiltrates and almost complete absence of tissue edema [85]. In a TNBS-induced colitis rat model, a combination therapy of AdSCs and sulfasalazine could attenuate the lesions via the S1P pathway [86]. In a TNBS- and DSS-induced colitis mice model, AdSCs were found to induce immunomodulatory macrophages and to protect them from experimental colitis and sepsis [34].

In a DSS-induced colitis mice model, AdSCs were demonstrated to ameliorate colitis by the suppression of inflammasome formation and regulation of M1 macrophage population through prostaglandin E2 [87]. In a DSS-induced colitis mice model, production of TSG-6 by AdSCs could ameliorate colitis through M2 macrophage induction [88]. In a DSS-induced colitis mice model, expression of PGE2 by AdSCs upregulated the expression of FOXP3+ Treg cells within the inflamed colonic tissue, which further dampened the inflammation and resolved the colitis injuries [89]. In a DSS-induced colitis mice model, the conditioned medium of AdSCs had immunomodulatory properties and the potential to reduce colitis inflammatory responses [90].

In a DSS-induced colitis mice model, the administration of AdSCs could alleviate colitis lesions by inhibiting the inflammatory and autoimmune responses with no adverse events [91]. In a DSS-induced colitis mice model, AdSCs were illustrated to ameliorate colitis lesions via down-regulating the expression of pro-inflammatory cytokines and suppressing the mucosal immune responses [92]. In a DSS-induced colitis mice model, injection of filtrated AdSCs could suppress histological inflammation, decrease the gene expression of inflammatory mediators, maintain the expression of tight junction proteins in the colon, present anti-apoptotic effects, and significantly improve the weight loss [36].

### 1.12. In Vivo Studies Using UCSCs in the Treatment of UC

In a DSS-induced colitis mice model, the use of UCSCs (1 × 10^6^ cells) and their exosomes (200 µg) showed a certain degree of immunosuppressive and improved therapeutic effects [93]. In a DSS-induced colitis mice model, the administration of UCSCs could alleviate the colonic lesions via the positive role of interferon-γ-mediated secretion of tryptophanyl-tRNA synthetases [94]. In a DSS-induced colitis mice model, a single-injection of UCSCs was shown to improve colitis lesions and to decrease the progression of acute colitis to chronic colitis [5]. Chang et al., in a DSS-induced mice colitis model, found that UCSCs injected subcutaneously and intraperitoneally could reduce and decrease the progression to chronic colitis. The anti-inflammatory effects of UCSCs were more prominent in the subcutaneous route in comparison to the intraperitoneal injection [95]. In a DSS-induced mice colitis model, the evaluation of the therapeutic efficacy of IL-1β-primed UCSCs showed an enhanced immunosuppressive capacity and ability for migration to the inflammatory site of the gut via the upregulation of chemokine receptor type 4 (CXCR4) expression [96]. In a DSS-induced mice colitis model, identical results were documented by Yang et al. for the crosstalk between UCSCs and T cells mediated by PGE2. Moreover, Yang et al. demonstrated an antiapoptotic influence through inducing the ERK pathway at the early stage of colitis development and inhibition of TNFα and IL-2, while promoting IL-10 in T cells [97]. EVs derived from UCSCs injected via tail vein were found to alleviate colitis in a mice model of colitis [98]. In a DSS-induced mice colitis model, UCSCs were proven to have a direct preventive effect other than the T-cell immunomodulatory properties, which were already known, suggesting that they are potential therapeutic targets in colitis treatment [99]. Human UCSCs were shown to reduce colitis in a DSS-induced mice colitis model by activating NOD2 signaling to COX2 [100]. Human UCSC-mediated modulation of IL-23/IL-17 could regulate inflammatory reactions and ameliorate colitis injuries [101] (Table 6).

### 1.13. In Vivo Studies on Treatment of UC Using Other Sources of MSCs

Injection of extracellular vesicles (EVs) derived from placental MSCs in a TNBS-induced colitis mice model revealed alleviation in colitis lesions by inhibiting the inflammation and oxidative stress pathways [102]. Transplantation of amnion-derived MSCs (AMSCs) and conditioned medium (CM) via enema in a TNBS-induced colitis rat model provided a significant improvement in endoscopic score, a significant decrease in infiltration of neutrophils and monocytes/macrophages, and a decrease in expression levels of TNF-α, CXCL1, and CCL2, leading to a significant improvement in colitis injuries [43]. Cell therapy with CM derived from intestinal MSCs in a TNBS- and DSS-induced colitis rat model showed that MSCs can produce pleiotropic gut trophic factors as a therapeutic potential in the treatment of colitis [37]. CM derived from amniotic fluid stem cells (AFSCs) in a DSS-induced colitis mice model demonstrated a successful amelioration of colitis lesions [44]. The therapeutic effect of EndSCs in a DSS-induced colitis mice model was shown via the downregulation of dickkopf-1 and the immunoregulating properties of these cells [103]. Transplantation of intestinal stem cells (ISCs) in a DSS-induced colitis mice model demonstrated the efficacy of cell transplantation in colitis injuries. The transplanted cells were shown to integrate into the colon, to cover the lacked epithelium, and to self-renew the crypts, leading to a functionally and histologically normal tissue [42]. Tonsil-derived stem cells (TSCs) and CM derived from TSCs illustrated equivalent effects in the improvement of inflammation in a DSS-induced colitis mice model. After cell therapy, weight gain, recovery of colon length, reduction of disease activity index, and decrease in the expression level of the proinflammatory cytokines, interleukin (IL)-1β, IL-6, and IL-17, were visible [45] (Table 7).

### 1.14. Limitations in the Treatment of UC by MSCs

There are some limitations in the treatment of UC using MSCs which are worthy of consideration. The different observable therapeutic results using MSCs in UC might be due to the variation in cell transplantation sources, cell transplantation number, modalities of cell administration, timing of infusions and the intervals between injections [104]. The in vivo study designs varied widely among different studies, including the type of colitis induction (DSS, TNBS, acetic acid, etc.), the animal model, the cell transplantation sources, the cell transplantation number, the modalities of cell administration, the timing of infusions, the intervals between injections, the assessment methods, and the follow-up time. It should also be noted that other factors can affect the outcome, as different laboratories use diverse methods to isolate and purify various MSCs, and accordingly, it is critical to define and standardize highly effective methods for MSCs yields. The mode of storage for MSCs is another variable influencing the results, while the storage is different in various laboratories regarding the cold chain, lyophilization, and transportation. Additionally, the sample size has an important role in in vivo studies and clinical trials in the treatment of patients suffering from UC. A low methodological quality may lead to a different outcome, even when no histopathologic or other direct indicators were used to estimate the role of MSCs (e.g., endoscope and MRI) in human studies.

In conclusion, the findings from in vivo studies and clinical trial are encouraging, suggesting that MSCs from various sources appear to be potentially safe and have anti-inflammatory and immunomodulating activities to target regenerative medicine and personalized therapy in UC patients. The information obtained from ongoing and future studies may result in a revolution in the management and treatment of UC. There is still a need for more prospective studies based on UC registries to apply standardized timescales, methodology, and definitions to enable better comparisons of UC patients across the world.

## Figures and Tables

**Table 1 jcm-09-03922-t001:** Undertaken case reports in the treatment of ulcerative colitis (UC) by various sources of MSCs.

Type of Study	Stem Cell Source (n)	Outcome	Reference
Case report	BMSCs (4.0 × 10^8^ cells, volume: 50 mL, submucosal injection)	Alleviation of UC symptoms, recovering from hematochezia, decrease in C reactive protein and lesions, bleeding, and edema in the sigmoid colon and rectum.	[53]
Case report	BMSCs (NA)	The first IV systemic cell transplantation and successful treatment of UC.	[54]
Case series	BMSCs (NA)	Improvement in clinical course, increase in duration of remission, and decrease in the risk of relapse.	[55]
Case series	BMSCs (NA)	Improvement in clinical course and decline in risk of relapse and frequency of hospital admissions after 2 years of follow-up.	[56]
Case series	BMSCs (NA)	Age as a prognostic factor in effective treatment of patients; in the elderly, the transplantation was less effective.	[57]
Case report	HSCs (NA)	Complete resolution of a severe form of UC.	[59]
Case series	BMSCs (NA)	Allogeneic BMSCs were safe and contributed to clinical improvement in patients with refractory UC.	[60]

MSCs: Mesenchymal stem cells. BMSCs: Bone marrow-derived stem cells. HSCs: Hematopoietic stem cells. NA: Not available. IV: Intravenously.

**Table 2 jcm-09-03922-t002:** Undertaken clinical trials in the treatment of UC by various sources of MSCs.

Type of Study	Stem Cell Source (n)	Outcome	Reference
Clinical trial	PBMCs (NA)	The CD4+IL-25R+ cells and LGR5+IL-25R+ cells significantly increased in the colonic mucosa of UC patients. Concomitant use of stem cells and IL-25 enhanced intestinal epithelial cell regeneration and healing.	[61]
Clinical trial, phase I/II	UCSCs (2.3–4.7 × 10^7^ cells/kg, 50 mL volume, IV)	Positive therapeutic effect of cell transplantation	[62]

MSCs: Mesenchymal stem cells. PBMCs: peripheral blood mononuclear cells. UCSCs: Umbilical cord-derived stem cells. NA: Not available. IV: Intravenously.

**Table 3 jcm-09-03922-t003:** Registered phase I/II clinical trials in the treatment of UC by MSCs.

ClinicalTrials.gov Identifier	Study	Number of Patients	Type of Stem Cell
NCT03299413	Hanan Jafar in University of Jordan	20	WJSCs
NCT02874365	Emmanuel Mas in Toulouse University Hospital, France	120	ISCs
NCT03115749	Guillaume Pineton in Montpellier University Hospital	60	NA
NCT03369353	Leslie Kean in Seattle Children’s Hospital, USA	625	HSCs
NCT03609905	Peng Yan in Liaocheng People’s Hospital, China	50	AdSCs
NCT 01914887	In Instituto de Investigaci on Hospital Universitario La Paz, Bolivia	8	AdSCs (60 × 10^6^ cells/kg, endoscopic injection in colonic submucosa)
NCT 01221428	In Qingdao University, China	NA	UCSCs (2 × 10^7^ cells/kg, 1 week later 1 × 10^7^ cells/kg, IV)
NCT 02442037	Affiliated Hospital to Academy of Military Medical Sciences, China	30	UCSCs (1 × 10^6^ cells/kg, IV, 3 weekly)
NCT 02150551	Children’s National Medical Center, Washington, USA	NA	BMSCs (1 × 10^6^ cells/kg, IV, 8 weekly)

MSCs: Mesenchymal stem cells. WJSCs: Wharton Jelly stem cells. ISCs: Intestinal stem cells. HSCs: Hematopoietic stem cells. AdSCs: Adipose-derived stem cells. UCSCs: Umbilical cord-derived stem cells. BMSCs: Bone marrow-derived stem cells. NA: Not available. IV: Intravenously.

**Table 4 jcm-09-03922-t004:** In vivo studies in the treatment of UC using BMSCs.

Method of UC Induction	Animal Model	Outcome	Reference
TNBS	Mice	Repair of injured intestinal mucosa.	[65]
TNBS	Rat	Healing in colonic lesions.	[66]
TNBS	Guinea-pig	Prevention of enteric neuropathic inflammation.	[67]
TNBS and DSS	Mice	Diminishing mucosal damage.	[68]
TNBS and DSS	Rat	Enhancement in the repair of an injured intestinal epithelium.	[37]
DSS	Rat	Ameliorating colonic lesions.	[69]
DSS	Rat	Repair of ulcerations.	[70]
DSS	Rat	Improvement in prompted colonic lesions.	[71]
DSS	Rat	Promoting the repair of colitis.	[72]
DSS	Mice	Preventive and fast recovery effects of mucosal damage.	[47]
DSS	Mice	Healing effects in colonic lesions.	[48]
DSS	Mice	Optimal mice colitis recovery.	[73]
DSS	Mice	Alleviation of colonic pathology.	[74]
DSS	Mice	Successful colonic mucosal regeneration.	[27]
DSS	Mice	Beneficial effects in healing of colonic lesions.	[75]
DSS	Mice	Attenuating UC.	[76]
DSS	Mice	No significant histopathologic or clinical improvement, limited therapeutic approach.	[77]
DSS	Mice	Promotion of tissue repair.	[78]
DSS	Mice	Reduction in intestinal inflammation.	[79]
DSS	Mice	Promoting healing effects in colonic lesions.	[80]
DSS	Mice	Enhancing repair in injured tissue.	[81]
DSS	Mice	Restoring intestinal mucosal permeability.	[82]
DSS	Mice	Sustained protection against acute inflammation in the long term.	[83]
Acetic acid	Rat	Ameliorating clinical manifestations and inflammation in UC.	[46]

TNBS: Trinitrobenzene sulfonic acid; DSS: Dextran sulfate sodium-induced colitis; MSCs: Mesenchymal stem cells; BMSCs: Bone marrow-derived stem cells.

**Table 5 jcm-09-03922-t005:** In vivo studies in the treatment of UC using AdSCs.

Method of UC Induction	Animal Model	Outcome	Reference
TNBS	Rat	Ameliorating colitis and preventing stenosis.	[46]
TNBS	Rat	Ameliorating colitis injuries with fewer inflammatory infiltrates in the absence of tissue edema.	[85]
TNBS	Rat	Attenuating colitis injuries.	[86]
TNBS and DSS	Mice	Protection from colitis injuries and sepsis.	[34]
DSS	Rat	Ameliorating colitis lesions.	[87]
DSS	Mice	Ameliorating colitis injuries.	[88]
DSS	Mice	Alleviation of colitis lesions.	[89]
DSS	Mice	The potential to reduce colitis inflammatory responses.	[90]
DSS	Mice	Alleviation of colitis injuries.	[91]
DSS	Mice	Amelioration of colitis lesions.	[92]
DSS	Mice	Suppressing histological inflammation and showing significant improvement in weight loss.	[36]

TNBS: Trinitrobenzene sulfonic acid; DSS: Dextran sulfate sodium-induced colitis; AdSCs: Adipose-derived stem cells; UC: Ulcerative colitis.

**Table 6 jcm-09-03922-t006:** In vivo studies in the treatment of UC using UCSCs.

Method of UC Induction	Animal Model	Outcome	Reference
DSS	Mice	Immunosuppressive and improved therapeutic effects in colitis injuries.	[93]
DSS	Mice	Alleviating colonic lesions.	[94]
DSS	Mice	Improving colitis lesions and decreasing the progression of acute colitis to chronic colitis.	[7]
DSS	Mice	Reducing and decreasing the progression to chronic colitis.	[95]
DSS	Mice	Therapeutic efficacy of cell transplantation and enhanced immunosuppressive ability.	[96]
DSS	Mice	Therapeutic effect of cell therapy on colitis injuries	[97]
DSS	Mice	Relieving colitis lesions.	[98]
DSS	Mice	A direct preventive effect and potential therapeutic targets in colitis treatment.	[99]
DSS and TNBS	Mice	Reducing colitis lesions.	[100]
TNBS	Mice	Ameliorating colitis injuries.	[101]

DSS: Dextran sulfate sodium-induced colitis; TNBS: Trinitrobenzene sulfonic acid; UCSCs: Umbilical cord-derived stem cells; UC: Ulcerative colitis.

**Table 7 jcm-09-03922-t007:** In vivo studies in the treatment of UC using other sources of stem cells.

Method of UC Induction	Animal Model	Type of Stem Cell	Outcome	Reference
TNBS	Mice	Placental stem cells	Alleviating colitis lesions.	[102]
TNBS	Rat	AMSCs	A significant improvement in colitis injuries.	[43]
TNBS and DSS	Rat	ISCs	Stem cells producedpleiotropic gut trophic factors to be a therapeuticpotential in the treatment of colitis.	[37]
DSS	Mice	AMSCs	Therapeutic potential of cell transplantation in colitis.	[44]
DSS	Mice	EndSCs	Therapeutic efficacy in colitis.	[103]
DSS	Mice	ISCs	Effective treatment of colitis injuries.	[42]
DSS	Mice	TSCs	The efficacy of cell therapy in colitis and improvement of inflammation.	[45]

UC: Ulcerative colitis; DSS: Dextran sulfate sodium-induced colitis; TNBS: Trinitrobenzene sulfonic acid; AMSCs: Amnion-derived stem cells; ISCs: Intestinal stem cells; AMSCs: Amniotic fluid mesenchymal stem cells; EndSCs: Endometrial stemcells; TSCs: Tonsil-derived stem cells.

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
