# Peer review of "Therapeutic Effect of Mesenchymal Stem Cells in Ulcerative Colitis: A Review on Achievements and Challenges"

_jcm, 2020, doi:10.3390/jcm9123922_

Round 1

Reviewer 1 Report

General Comments

It is clear that this is not a finalized manuscript.  Only fully completed manuscripts should be submitted.

There are numerous problems with the use of the English language.

There are excessive references to one of the authors, Mehrabani.

The author’s discussion of MSCs, what they are, where they are derived and limitations on their use needs to be enhanced.

Specific Comments

Case reports should not be included in this type of review since they only involve a small number of individuals. 

If absolutely necessary, they can be aggregated and assessed in 1-2 sentences.

L255, injection of MSCs.  This needs further information as to sites of injection, dosage, source of MSCs.

Reviewer 2 Report

In the present work, the authors describe the effects of mesenchymal stem cells (MSC) from different tissues and origins in the indications of ulcertaiver colitis.

This is an interesting well documented paper. Nevertheless, the main problem of this work is a bad referecing of literature studies. For example, l53 , ref (8) doees not correspond to the right one.; line 17, ref (17) is not relevant. in fact, there are some sections with wrong refernces and some others with the correct ones. It is too long to describe, but this is a recurrent problem all along the paper. 

Another problem is that the paper is not easily readable, since there are redondant comments and results, which may be presented though a more "synthetic" and clear manner

So, my main comments are the following ones :

  1. major problems of references (see above)
  2. the authors should introduce the tissue origins and sources of MSC in a better and more documenetd way; in particular, the roles of source and origins in the therapeutic effects of MSC in general. Are their therpautic effects modified by their sources and origins , in geenral manner
  3. the authors should introduce and shortly describe the animal models of ulcerative colitis (UC)
    1. the parameters studied in the different reported studies should be clearly indicated (in particular those concerning the outcome  in tables (3 & 4, in particular) : endoscopic, histological ,etc...)
  4. The main mechanisms of action of MSC in UC should be more clearly highlighted and thios mùay avopid the redundancy within the diffreent paragraphs (BM-MSC, AdMSC, UCMSC, etc...). 
  5. l 222 : table 3 should be indicated rather than table 1
  6.  a list of abbreviation may be proposed

Reviewer 3 Report

The review has been well written. However, the authors should consider these suggestions:

  • in vivo studies should be described before the clinical trials
  • a specific paragraph about the secretome isolated from MSCs and their clinical benefits should be insert 
  • in the paragraph "Limitations in treatment of UC by MSCs"  a discussion about the dose, the GMP production and the costs of MSC on large scale for clinical use should be added 

In the file the authors could see some  minor comments
